# Diagnosis of Acute Aortic Syndromes on Non-Contrast CT Images with Radiomics-Based Machine Learning

**DOI:** 10.3390/biology12030337

**Published:** 2023-02-21

**Authors:** Zhuangxuan Ma, Liang Jin, Lukai Zhang, Yuling Yang, Yilin Tang, Pan Gao, Yingli Sun, Ming Li

**Affiliations:** 1Radiology Department, Huadong Hospital, Affiliated with Fudan University, Shanghai 200040, China; 2Radiology Department, Huashan Hospital, Affiliated with Fudan University, Shanghai 200040, China; 3Institute of Functional and Molecular Medical Imaging, Shanghai 200040, China

**Keywords:** radiomics, acute aortic syndromes, non-contrast CT

## Abstract

**Simple Summary:**

Computed tomography angiography can provide sufficient information for the diagnosis of acute aortic syndromes. However, non-contrast computed tomography images in the emergency department, compared with CTA, are more easily accessible and convenient and have lower radiation doses with fewer contraindications. We retrospectively analyzed 325 patients’ non-contrast CT images from 2 independent medical centers and established an acute aortic syndrome recognition model based on the radiological features of non-contrast CT images through feature extraction and screening. This model can effectively detect acute aortic syndrome on non-contrast CT images with high sensitivity, AUC, and robustness. More importantly, it can diagnose patients who do not have specific imaging findings on non-contrast CT images. It has important clinical applications for the screening of acute aortic syndrome, especially in the emergency department.

**Abstract:**

We aimed to detect acute aortic syndromes (AAS) on non-contrast computed tomography (NCCT) images using a radiomics-based machine learning model. A total of 325 patients who underwent aortic CT angiography (CTA) were enrolled retrospectively from 2 medical centers in China to form the internal cohort (230 patients, 60 patients with AAS) and the external testing cohort (95 patients with AAS). The internal cohort was divided into the training cohort (*n* = 135), validation cohort (*n* = 49), and internal testing cohort (*n* = 46). The aortic mask was manually delineated on NCCT by a radiologist. Least Absolute Shrinkage and Selection Operator regression (LASSO) was used to filter out nine feature parameters; the Support Vector Machine (SVM) model showed the best performance. In the training and validation cohorts, the SVM model had an area under the curve (AUC) of 0.993 (95% CI, 0.965–1); accuracy (ACC), 0.946 (95% CI, 0.877–1); sensitivity, 0.9 (95% CI, 0.696–1); and specificity, 0.964 (95% CI, 0.903–1). In the internal testing cohort, the SVM model had an AUC of 0.997 (95% CI, 0.992–1); ACC, 0.957 (95% CI, 0.945–0.988); sensitivity, 0.889 (95% CI, 0.888–0.889); and specificity, 0.973 (95% CI, 0.959–1). In the external testing cohort, the ACC was 0.991 (95% CI, 0.937–1). This model can detect AAS on NCCT, reducing misdiagnosis and improving examinations and prognosis.

## 1. Introduction

Acute aortic syndrome (AAS) includes classic acute aortic dissection (AAD), intramural hematoma (IMH), penetrating atherosclerotic aortic ulcer (PAU), and limited intimal tear (LIT) [1]. These conditions share common pathophysiological pathways (e.g., breakdown of the intima and media), clinical characteristics, and diagnostic and therapeutic challenges [2]. A recent American study found an incidence of 7.7 per 100,000 person-years for AAD, IMH, and PAU, in which the incidence of AAD was 4.4 per 100,000 person-years, whereas those of PAU and IMH were lower [3]. AAS is a life-threatening disease with a low prevalence, and it frequently presents with nonspecific clinical symptoms and lacks specific biomarkers [4]. Computed tomography angiography (CTA) provides a complete and detailed anatomy of the entire aorta and its branches. Importantly, it is the most frequently used modality for the initial assessment of AAS [5,6,7].

However, the use of iodinated contrast media (ICM) leads to a non-negligible risk of hypersensitivity reactions. Immediate and nonimmediate hypersensitivity reactions to ICM have been reported at a frequency of about 0.5–3% in patients receiving nonionic ICM [8]. Compared with CTA, non-contrast computed tomography (NCCT) scans in the emergency department (ED) are more easily accessible and convenient and have lower radiation doses with fewer contraindications [9,10,11]. However, AAS diagnosis on NCCT images without significant characteristics (e.g., displaced calcified intimal flaps, intraluminal linear high density, intramural hematoma, and aneurysmal dilatation) is difficult, and NCCT has lower sensitivity even when the findings are interpreted by experienced radiologists [12]. Radiomics is a quantitative approach to medical imaging that can extract a large number of predefined high-throughput features, followed by statistical methods to filter the features most relevant to the results [13,14].

The radiomics features include the shape, intensity, and texture features of the original image of the lesion, as well as the image features transformed by multiple filters such as wavelet and Gaussian Laplace (LoG). By combining feature selection methods and machine learning algorithms, the prediction models can be constructed on a training cohort and further evaluated on a test cohort. This technique has been widely used in oncology studies, but rarely for vascular diseases [15]. This study aimed to explore the unique radiomics features from NCCT images of patients with AAS and use these features to accurately identify AAS on NCCT images.

In our study, we attempted to propose a machine learning model based on radiomics features, which could effectively detect AAS on NCCT images. We further validated and compared the assay performance of two radiologists at two independent centers. We expect that the radiomics-based methods, combined with the features extracted from NCCT images, can improve the sensitivity of AAS prediction.

## 2. Methods

### 2.1. Study Design and Population

This retrospective study was approved by the Institutional Review Board of our hospital (Approval No. 20220047) and was conducted according to the tenets of the Declaration of Helsinki. The requirement for informed consent was waived owing to the retrospective nature of the study.

This study initially evaluated 2251 consecutive patients who underwent CTA in our hospital between January 2010 and March 2022. AAS was confirmed through CTA interpretation by radiologists. In total, 299 patients were diagnosed with AAS (285, 5, and 9 patients were diagnosed with AAD, IMH, and PAU, respectively), while 1952 patients were diagnosed without AAS. Among them, patients with both NCCT images and CTA images before surgical repair and patients diagnosed with AAS were included. We excluded low-quality images from patients with prosthetic vessels and stents, who were readily diagnosed by a human expert on the NCCT. Ultimately, we excluded these patients as well as those with incomplete images. Finally, 60 (50, 3, and 7 patients who were diagnosed with AAD, IMH, and PAU, respectively)/299 patients with AAS and 170/1952 propensity-matched patients without AAS were included. Further, 95 patients (90, 1, and 4 patients with AAD, IMH, and PAU, respectively) diagnosed with AAS at another medical center between January 2021 and March 2022 were included as the external validation dataset (Figure 1).

### 2.2. CT Image Acquisition

All CT scans were performed using post-64-detector row CT scanners from Siemens (Somatom Definition Flash, Somatom Force, or Somatom Drive, Forchheim, Germany) and GE (Revolution CT, Discovery CT750 HD, or 64-slice LightSpeedVCT, GE Medical Systems, Milwaukee, WI, USA). Every scan started with non-contrast scanning from the thoracic inlet to the pubic symphysis to cover the entire aorta. Subsequently, contrast-enhanced CT was performed over the same area during the systemic arterial phase. The slice thickness was 1–5 mm for NCCT images and 1–1.5 mm for contrast-enhanced CTA images. The other scanning parameters were as follows: rotation time, 0.5 s; pitch, 1.2–1.375; matrix, 512 × 512; standard resolution algorithms; and tube voltage, 80–100 kV (Somatom Definition Flash or Somatom Force or Somatom Drive, Germany) and 120 kVp (Revolution CT or Discovery CT750 HD or 64-slice LightSpeedVCT, GE Healthcare, USA). The tube current was adjusted automatically using the noise index mode.

### 2.3. Aorta Segmentation and Radiomic Feature Extraction

Open-source software (3D Slicer, version 4.13.0; National Institutes of Health; https://www.slicer.org; accessed on 7 August 2021) was used by two radiologists to manually delineate the volume of interest (VOI) of the 325 aortas at the voxel level. A total of 1218 radiomic features were extracted through Pyradiomics (version: 3.0.1). The main radiomic features were first-order statistics (18 features); shape statistics (14 features); texture features including gray-level co-occurrence matrix (GLCM), gray-level size zone matrix (GLSZM), gray-level dependence matrix (GLDM), gray-level run length matrix (GLRLM), 68 features; statistical features derived from texture matrices in LoG filtered domain (1.0–5.0 mm kernels), 430 features; and statistical features derived from texture matrices in wavelet filtered domains, 688 features.

### 2.4. Consistency of Segmentation and Radiomics Features

Dice and 95%Hausdorff distance (HD95) were used to evaluate the differences in aortic segmentation between radiologists [16]. HD95 is a metric using hausdorff distance to measure the 95% quantile of the surface distance. HD95 was used to evaluate the degree of matching between manually segmented images and ground truth, and the unit of measurement was mm. The larger the HD95, the higher the mismatch between the two images [17]. Intraclass correlation coefficient (ICC) was used to evaluate the reliability of radiomics values between the two radiologists [18]. ICC is the reliability coefficient measuring and evaluating inter-observer reliability and test-retest reliability. In our study, ICC was calculated based on a single-measurement, absolute-agreement, two-way random-effects model. Originally, all VOI segmentations in the 325 patients were performed by one radiologist (radiologist A, with >8 years of experience in vascular CT interpretation). Then, 30 random CT images obtained by another radiologist (with >10 years of experience in cardiovascular radiology) were selected for the evaluation of reliability.

### 2.5. Radiomic Signature Construction

The radiomic signature was constructed via a limestone combination of nine selected features weighted by their respective coefficients. Scikit-learn (version 1.1.1) is a Python module for machine learning built on top of SciPy. We used sklearn to complete algorithms such as data preprocessing, classification prediction model, and model evaluation. First, Preprocessing-StandardScaler was used to normalize the extracted radiomic features. Thus, 230 patients who were enrolled from our hospital were randomly divided into the training cohort (59%, 135 patients [36 patients with AAS and 99 patients without AAS]), validation cohort (21%, 49 patients [15 patients with AAS and 34 patients without AAS]), and internal testing cohort (20%, 46 patients [9 patients with AAS and 37 patients without AAS]). Then, three main algorithms, namely, Logistic Regression (LG), Decision Trees (DT), Support Vector Machine (SVM), and K-nearest-neighbor (KNN), were selected for data training. For the training data set, a series of base learners was trained and combined through combination strategies to form a strong learner to achieve the purpose of learning from others. The following ensemble learning algorithms were used: eXtreme Gradient Boosting (XGBoost) [19], Gradient Boosting Decision Tree (GBDT), and Random Forests (RF). XGBoost and GBDT are all based on the Boosting framework in ensemble learning, while RF belongs to the Bagging algorithm in ensemble learning. The Gaussian Discriminant Analysis model as a generative learning algorithm was also used. Scikit-learn also includes three classes of Naive Bayes classification algorithms: GaussianNB (GNB), MultinomialNB, and BernoulliNB. GNB, a Naive Bayes with Gaussian distribution prior, was selected in this study to train the data. The models were built in the training cohort and evaluated in the internal and external testing cohorts.

### 2.6. Radiologists–Model Collaboration

Radiologists A (junior radiologist with 5 years of experience in chest imaging interpretation) and B (senior radiologist with over 20 years of experience in chest imaging interpretation) interpreted the occurrence of AAS on the NCCT images of the external testing set. All interpretations were performed independently and in a double-blinded manner. Then, radiologists A and B completed the case diagnosis together. Two months later, radiologist B, assisted with the radiomics-based model, interpreted the AAS on the NCCT images of the same external testing set in a radiologist–model collaboration.

## 3. Statistical Analysis

All statistical analyses were performed using R (version 3.6.3), Python (version 3.9.7), and SPSS (version 22). Kydavra (version 0.3.1) is a Python scikit-learning-inspired package for feature selection. The following methods of feature selection from Kydavra were used: Mann–Whitney U test selector, Pearson correlation selector, MUSE selector, and Least Absolute Shrinkage and Selection Operator (LASSO) selector. Significant results were defined as those with *p*-values less than 0.05. A receiver operating characteristic curve was constructed, and the detection efficiency of the model was evaluated based on four aspects: AUC, ACC, sensitivity, and specificity. To select a stable model parameter, we set the epoch parameter to 1000 and our training environment was completed in a graphics processing unit (GPU, RTX 3090) environment. Unlike deep learning, the classification model based on radiomics data does not require the support of a GPU; our model can also run in a central processing unit (CPU) environment. The median was selected to represent the monitoring performance of the model to avoid being affected by the maximum or minimum value of the distribution sequence of the results.

## 4. Results

### 4.1. Patient Characteristics

A total of 230 patients included from our hospital as the internal set were randomly divided into a training cohort (59%, 135 patients [36 patients with AAS and 99 patients without AAS]), validation cohort (21%, 49 patients [15 patients with AAS and 34 patients without AAS]), and internal testing cohort (20%, 46 patients [9 patients with AAS and 37 patients without AAS]). In addition, another 95 patients with AAS were recruited as an external validation set from another center. Table 1 shows the clinicodemographic patient characteristics and CT image parameters by cohort. The training and validation cohorts were combined because they were used to find the optimal hyperparameters and train the model. The total internal cohort consisted of the training, validation, and internal testing cohorts because they were used for comparison with the external testing cohort.

There were significant differences in the age and sex composition of the study cohort (*p* < 0.05). The majority of the patients in the current study had Stanford B-type dissection. No significant differences were observed in intimal calcification between the AAS and non-ASS (NASS) groups. Owing to different scanning methods and inspection purposes, the slice thickness of the image data we collected was significantly different between and within groups (*p* < 0.001), and the slice thickness of the external testing cohort was thicker than that of the other cohorts.

### 4.2. Agreement Analysis between Radiologists

A total of 1218 radiomics results were extracted from Pyradiomics. We calculated the Dice and 95%Hausdorff distance of 30 random patients out of the 325 patients; the median ± standard deviation of Dice was 0.932 ± 0.026, and the median ± standard deviation of HD95 was 0.101 ± 0.166 (specific results are shown in Table 2). Similarly, we also evaluated the radiomics of 30 patients between the two radiologists. Among all the 1218 radiomic features, 1056 features were found to have reliable inter- and intra-reader agreement for radiomic feature extraction (ICC > 0.75), detailed in Appendix A.

### 4.3. Radiomic Feature Selection

Among 1056 radiomic features, the Mann–Whitney U test selector (threshold value < 0.005) was used to compare the significant differences among 1056 features, and 542 features were retained by the Mann–Whitney U test. Next, we artificially set the parameters of the MUSE selector to reserve 30 features in the next step. Finally, nine features were selected through LASSO. The nine features with nonzero coefficients were selected using the LASSO LG algorithm on the basis of the training set (Figure 2 and Table 3). We analyzed the differences in the nine radiomics features between the AAS and NASS groups. There were significant differences in the radiomics features between the two groups, and all *p* values between the two groups were <0.05. Among the nine finally established radiomics features, the most important was original_shape_SurfaceVolumeRatio.

### 4.4. Comprehensive and Comparative Analysis of Model Performance in the Validation Cohort

The prediction performance of each model based on the internal datasets is presented in Table 4, Table 5, and Figure 3. Most of the classification models established with radiomics features showed good performance in the validation cohort. Among these, the SVM model performed the best. In the validation cohort, the SVM model had an AUC of 0.993 (95% CI, 0.965–1); ACC, 0.946 (95% CI, 0.877–1); sensitivity, 0.9 (95% CI: 0.696–1); and specificity, 0.964 (95% CI, 0.903–1). In the internal testing cohort, the SVM model had an AUC of 0.997 (95% CI, 0.992–1); ACC, 0.957 (95% CI, 0.945–0.988); sensitivity, 0.889 (95% CI, 0.888–0.889); and specificity, 0.973 (95% CI, 0.959–1). In the external validation cohort, the ACC was 0.991 (95% CI, 0.937–1). The performance of the KNN model and the decision tree model was not as good as that of the other models, in both the training and validation cohorts. In the external testing cohort, the ACC of KNN was only 0.505 (95% CI, 0.541–0.640), detailed in Appendix A.

### 4.5. Performance of the Detection Models and Radiologist in the External Testing Cohort

The prediction performance of each model in the external testing cohort is presented in Table 6. We analyzed the diagnosis of the external data by the two radiologists. If they accurately determined whether a patient had AAS, the result was considered a diagnosis. If the diagnosis was clearly missed, it was considered a missed diagnosis. However, if the diagnosis of AAS was suspected and the radiologists recommended further examination, the diagnosis was regarded as suspicious. As shown in Table 6, the occurrence of AAS on the NCCT images of the external testing cohort was interpreted independently by one radiologist, and the rate of missed diagnosis of AAS was 0.189. Six patients with AAS were missed despite a review by senior radiologists. Our SVM radiomics model was able to accurately detect AAS patients in the external testing cohort. Similarly, patients with AAS who presented as nonspecific on the NCCT were more likely to be identified when radiologists used our radiomics SVM model to assist in diagnosis.

Figure 4 shows the NCCT and CTA images of six patients with AAS. Characteristic features (e.g., shifted calcified intimal valves, linear intraluminal hyperdensity, intramural hematoma, and aneurysmal dilatation) are difficult for human experts to detect on NCCT images. These features are very evident on CTA and are represented by red circles in the figure. None of the four radiologists made an accurate diagnosis on the NCCT, and all six patients were missed. Figure 5, respectively, shows the radiomic feature values of the six patients. Among the nine radiomic features ultimately established, we observed four features with the top four important coefficients. As clearly shown, the characteristic values of these six patients were closer to the mean values in the AAS group.

## 5. Discussion

CTA is the gold standard modality for acute diagnosis of AAS. However, the appropriate selection of patients requiring CTA is a primary diagnostic challenge in the ED. Moreover, the use of ICM leads to a non-negligible risk of hypersensitivity reactions. Given that most patients with AAS-compatible symptoms may be afraid of more prevalent conditions (e.g., musculoskeletal pain, gastrointestinal disease, coronary artery disease, primary stroke, and syncope), and accurate biomarkers are not available, NCCT, which is convenient and affordable, can be performed for screening in the ED. In clinical practice, it is of great significance to make full use of NCCT scans to assist in the early warning of AAS, and it may greatly improve the outcomes of patients.

Radiomics feature-based models have successfully achieved various tumor-related predictions but are rarely used in the analysis of vascular diseases. Hata et al. [20] used a convolutional neural network model to detect the occurrence of AD on 2D NCCT images. They trained the model using 6688 NCCT images (slice thickness, 5 mm) from 170 patients (85 patients with AAD, 85 patients without AAD) in a single center. The AUC of the developed algorithm was 0.940, and the ACC was 90.0%; sensitivity, 91.8%; and specificity, 88.2%. Yi et al. [12] developed a combination model of both deep learning model scores and extracted morphological features from aortic segmentation. The ACC, sensitivity, and specificity of the model in the internal testing cohort were 0.897, 0.862, and 0.923, respectively, and reached 0.730, 0.978, and 0.554, respectively, in the external testing cohort. Xiong et al. [21] developed and evaluated a cascaded deep-learning framework that consisted of a 3D segmentation network and a synthesis network. A conditional generative adversarial network was used to map NCCT images to contrast-enhanced CT images of the aortic region non-linearly to assist physicians in diagnosis. A radiomics-based model or deep learning-based model using NCCT images for AD detection is feasible, and these artificial intelligence-powered approaches have been proven to increase the efficiency of the physician’s workflow, as they help in the detection of non-specific image features of AAS that could not be captured by human eyes. However, this has not been reported in detail in previous studies. The presence of significant characteristics (e.g., displaced calcified intimal flaps, intraluminal linear high density, intramural hematoma, and aneurysmal dilatation) on NCCT would suggest that this patient may have AAS. However, the intima of the aorta is not identifiable on NCCT in most cases. This explains why human experts also find it very hard to diagnose AAS with complete accuracy on NCCT images. In the present study, AAS diagnosis was missed by senior radiologist B in six patients, and we reviewed the NCCT images of these six patients (Figure 4 and Figure 5) without specific image features, such as displaced calcified intimal flaps, intraluminal linear high-density, intramural hematoma, and aneurysmal dilatation. Five patients with AAS were identified after combining findings from the radiologists and the model, and all six patients were successfully diagnosed with AAS. This indicated that our model could assist radiologists in diagnosing all kinds of AAS regardless of the presence or absence of specific imaging features, thus making our model valuable in clinical practice.

In our study, eight different novel machine learning algorithms were applied to construct detection models for AAS, and the SVM model outperformed the other prediction models in both training and test sets in our study. We demonstrated that a radiomics-based machine learning model could efficiently detect ASS on NCCT images with good robustness. The model has been validated using data sets from two independent clinical centers. Furthermore, our established model could identify the occurrence of AAS in patients without significant features on NCCT images. We identified nine radiomics features that could be used to detect AAS on NCCT scans. The top three most important factors in the results were original shape Surface Volume Ratio, wavelet LLH first-order mean, and original glrlm LRHGLE; shape statistics in the AAS diagnosis have important significance. In our study, Surface Volume Ratio and Wavelet LLH first-order mean were found to be lower in the AAS group. LRHGLE was found to be higher in the AAS group. This is consistent with our findings indicating that for the Surface Volume Ratio, a ratio of the surface area to the volume of the mask, a lower value indicates a more compact (sphere-like) shape, which may suggest that the part of the blood vessel in the AAS group in our study showed aneurysm-like dilatation. LRHGLRE measures the joint distribution of the long run lengths with higher gray-level values, with higher values indicating a greater proportion of high-density areas in the vessels, suggesting that the blood flow density was lower in the NAAS group than in the AAS group in this study. Other radiomics features were related to image homogeneity and heterogeneity. These findings indicate that the model is of great clinical significance to assist radiologists in AAS detection and early diagnosis. A radiomics-based model with a higher sensitivity for AAS is helpful for patients with an acute diagnosis, avoiding misdiagnosis and delayed diagnosis.

Reproducibility is one of the most problematic aspects of radiomics, and the core of it is the segmentation of ROI, which may directly or indirectly affect the establishment and accuracy of diagnostic models. Initial studies used image segmentation based on the largest slice of the lesion. Zhou et al. [22] conducted a single-center retrospective analysis of 50 patients with typical AD. The patients were selected based on the location of the aortic dissection on CTA and the largest lesion layer on NCCT images was selected to mark the corresponding ROI; 14 features were selected by extracting the radiomic features. The ACC, sensitivity, specificity, and AUC were 94.3% (66/70), 91.2% (31/34), 97.2% (35/36), and 0.988 (95% CI, 0.970–0.998), respectively. The corresponding values in the validation group were 90.0% (27/30), 94.1% (16/17), 84.6% (11/13), and 0.952 (95% CI, 0.883–0.986). Guo et al. [23] developed an NCCT-based radiomic signature to detect thoracic ADs, and the model achieved an AUC of 0.90 (95% CI, 0.82–0.98) in the external testing set. Our study is based on the VOI. It can comprehensively reflect the heterogeneity of the subjects and some potentially meaningful characteristics. This reinforces the importance of segmentation consistency. In this study, the radiomics study was based on the intact aortic VOI. Complete aortic segmentation is less affected by subjective analysis and professional level. It does not need to accurately outline the lesion level, but merely by delineating the complete aortic mask, which is less affected by the evaluator. At the same time, it can comprehensively and accurately evaluate and describe the quantitative characteristics of the observation object, to meet the requirements of precision medicine.

However, our study has several limitations. First, there may be a potential selection bias. The experimental subjects included in this study were all emergency patients who underwent NCCT and enhanced CT scans, and some asymptomatic AAS patients who had been missed after initial screening with NCCT images. Although our model can accurately detect AAS in images from two medical centers, more clinical information needs to be collected to validate the robustness of the model in cases where AAS is difficult to detect with the naked eye. Second, this was a retrospective study, the training sample size was small, more detailed clinical data were lacking, the image time span of the training set was large, and the image quality was not uniform. Further, images of AAD, IMH, PAU, and LIT can be difficult to fine grain. This study primarily aimed to assist radiologists in the diagnosis of AAS using only NCCT and avoid missed diagnoses. It is difficult for physicians to mark the lesions on NCCT accurately and completely, and this remains a challenge for accurate positioning and identification of the location of the aortic rupture. A multi-center prospective trial is needed to complete the accurate classification of AAS and accurate identification of the rupture.

## 6. Conclusions

Our proposed radiomics-based machine learning model could effectively detect the AAS on NCCT images with higher sensitivity, AUC, and good robustness, including on NCCT images without specific image features, it will add great value to the clinical practice in AAS screening, especially in ED.

## Figures and Tables

**Figure 1 biology-12-00337-f001:**
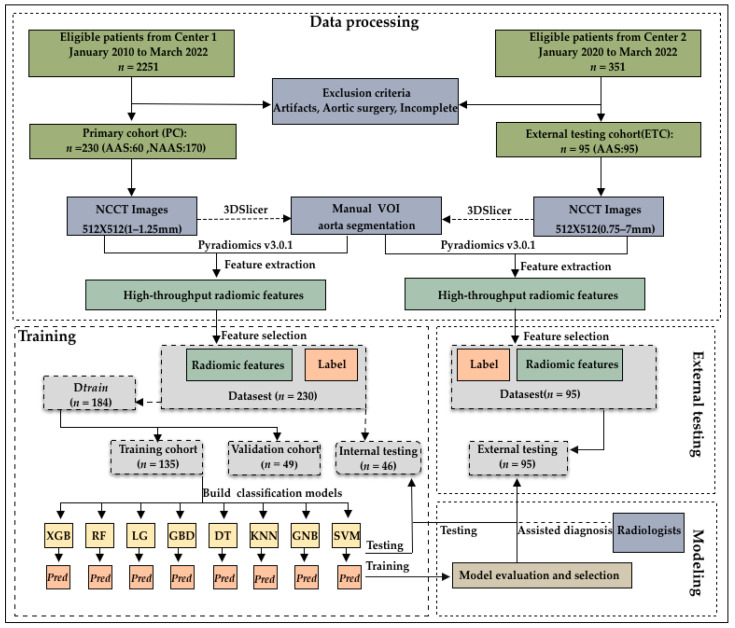
Flowchart of patient enrollment and model construction. The 325 patients enrolled were divided into a training cohort (*n* = 135), validation cohort (*n* = 49), internal testing cohort (*n* = 46), and external testing cohort (*n* = 95).

**Figure 2 biology-12-00337-f002:**
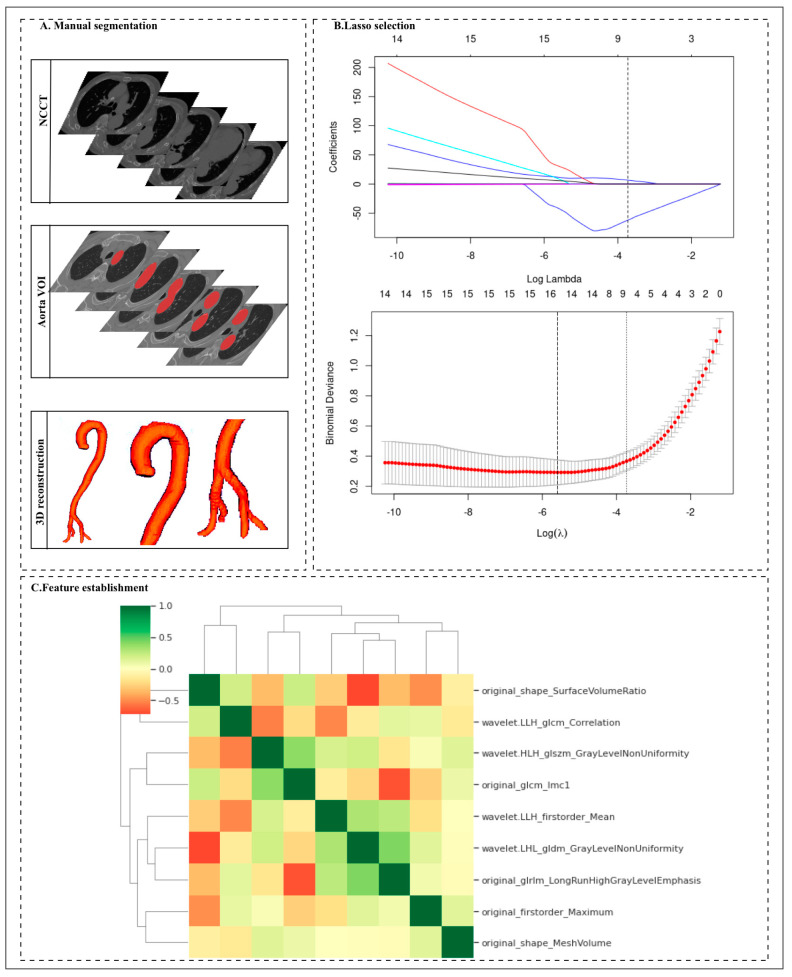
Workflow for the radiomic analysis. (**A**) The non-contrast computed tomography (NCCT) images were segmented by two radiologists to manually delineate the volume of interest (VOI) of the 325 aortas at the voxel level. Then, the radiomic features were extracted from the aorta mask through Pyradiomics. (**B**) Different color curves in the figure represent the change trajectory of each independent variable coefficient. The ordinate is the coefficient value, and the upper abscissa is the number of non-zero coefficients in the model at this time. (**C**) Nine radiomic features were selected by the LASSO logistic regression algorithm on the basis of the training cohort.

**Figure 3 biology-12-00337-f003:**
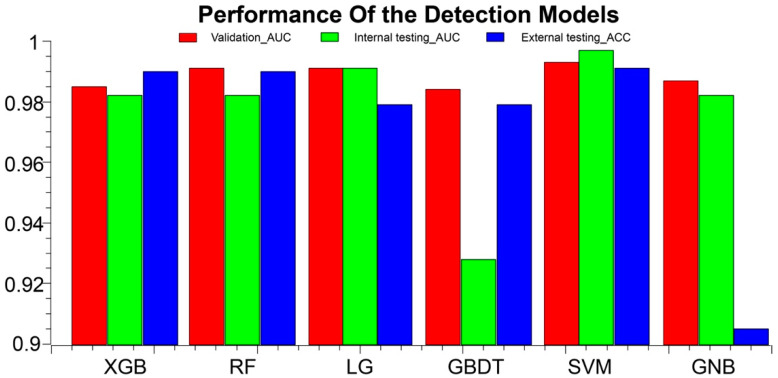
AAS detection performance results of the models in the training, validation, and external test cohorts. ACC, accuracy; AUC, area under the curve; XGB, eXtreme Gradient Boostin; RF, Random Forest; LG, logistic regression; GBDT, Gradient Boosting Decision Tree; SVM, Support Vector Machine; GNB, GaussianNB.

**Figure 4 biology-12-00337-f004:**
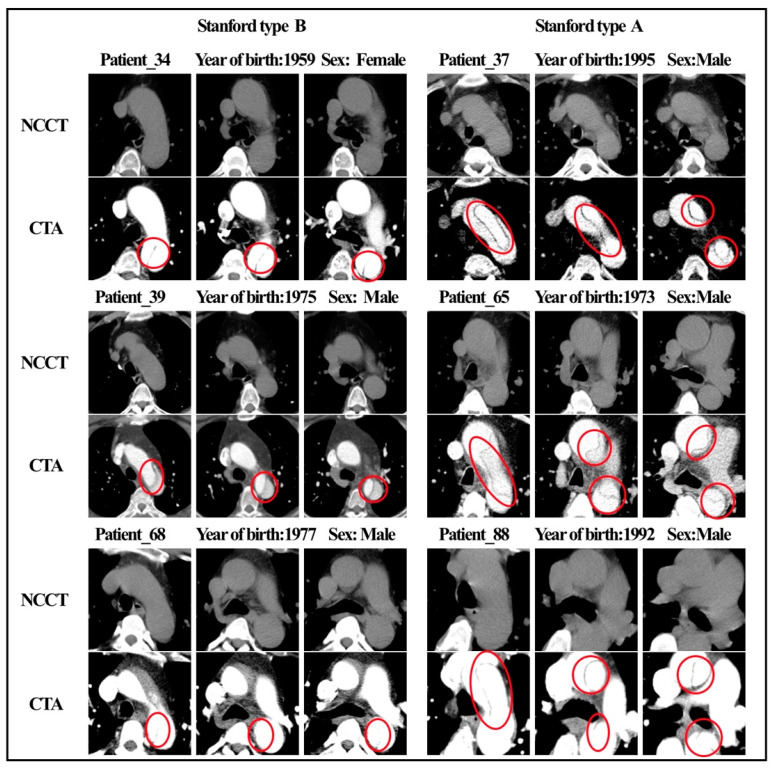
NCCT and CTA images of 6 patients with AAS.

**Figure 5 biology-12-00337-f005:**
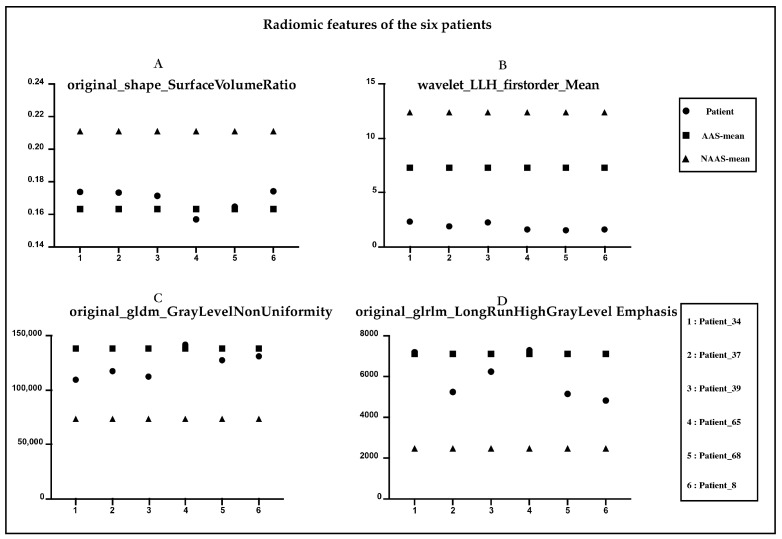
Radiomic features of the six patients diagnosed with AAS. (**A**) original_shape_Surface_Volume Ratio. (**B**) Wavelet_LLH_firstorder_Mean. (**C**) original_gldm_GrayLevelNonUniformity. (**D**) original_glrlm_LongRunHighGrayLevelEmphasis.

**Table 1 biology-12-00337-t001:** Clinical and imaging characteristics of patients in the training, validation, and external testing cohorts.

	Training and Validation Cohort			Internal Testing Cohort			Total Internal Cohort	External Testing Cohort	*p* Value
	AAS	NAAS	*p* Value	AAS	NAAS	*p* Value	AAS		
Number of patients	51	133	-	9	37	-	60	95	-
Age (years), (mean ± SD)	72.59 ± 14.61	66.15 ± 13.1	0.006	73.11 ± 16.40	66.03 ± 14.22	0.200	73.00 ± 14.28	65.24 ± 11.41	<0.001
Sex (n, %)			<0.001			<0.001			<0.001
Male	46 (90.2)	67 (50.4)		9 (100.0)	21 (56.8)		55 (1.6)	75 (79.0)	
Female	5 (9.8)	66 (49.6)		0 (0)	16 (43.2)		5 (8.4)	20 (21.0)	
Stanford type									
A (n, %)	10 (19.6)			3 (33.3)		<0.001	13 (21.7)	7 (7.4)	<0.001
B (n, %)	41 (80.4)			6 (66.7)			47 (78.3)	88 (92.6)	
Intimal calcification	28 (54.9)	63 (47.36)	0.371	4 (44.4)	17 (45.9)	0.777	32 (53.3)	30 (31.6)	0.020
Slice thickness (mm), (mean ± SD)	1.65 ± 1.27	1.25 ± 0	<0.001	1.13 ± 0.13	1.25 ± 0	<0.001	1.56 ± 0.99	5.00 ± 0	<0.001

The *p*-values are calculated using the Mann–Whitney U test or Pearson’s chi-squared test as appropriate.

**Table 2 biology-12-00337-t002:** Dice and 95%Hausdorff distance of 30 random cases out of the 325 patients.

Cases	Dice	HD95 (mm)	Case	Dice	HD95 (mm)
Case1	0.941	0.070	Case16	0.937	0.078
Case2	0.928	0.100	Case17	0.945	0.609
Case3	0.935	0.162	Case18	0.869	0.610
Case4	0.931	0.531	Case19	0.901	0.198
Case5	0.926	0.114	Case20	0.990	0.098
Case6	0.959	0.044	Case21	0.858	0.486
Case7	0.938	0.084	Case22	0.930	0.101
Case8	0.939	0.067	Case23	0.932	0.094
Case9	0.939	0.186	Case24	0.917	0.113
Case10	0.952	0.054	Case25	0.940	0.083
Case11	0.916	0.101	Case26	0.886	0.314
Case12	0.943	0.066	Case27	0.936	0.087
Case13	0.918	0.145	Case28	0.919	0.097
Case14	0.903	0.135	Case29	0.929	0.109
Case15	0.929	0.082	Case30	0.959	0.048
Median ± SD		Dice: 0.932 ± 0.026		HD95: 0.101 ± 0.166	

**Table 3 biology-12-00337-t003:** The most predictive subset of features and the important coefficients.

The Most Predictive Subset of Features	Important Coefficients	ICC	*p* Value
original_shape_MeshVolume	0.080	0.974	<0.05
original_shape_SurfaceVolumeRatio	0.365	0.890	
original_firstorder_Maximum	0.020	0.790	
original_glcm_Imc1	0.030	0.805	
original_glrlm_LongRunHighGrayLevelEmphasis	0.162	0.964	
original_gldm_GrayLevelNonUniformity	0.154	0.980	
Wavelet_LLH_firstorder_Mean	0.217	0.808	
Wavelet_LLH_glcm_Correlation	0.080	0.935	
Wavelet_LHL_glszm_GrayLevelNonUniformity	0.040	0.984	

ICC, intraclass coefficient.

**Table 4 biology-12-00337-t004:** Performance of the detection models in the validation cohort.

Models	Validation Cohort (N = 49)			
	ACC (95% CI)	Sensitivity (95% CI)	Specificity (95% CI)	AUC (95% CI)
XGB	0.919 (0.846, 1.00)	0.857 (0.616, 1.00)	0.963 (0.884, 1.00)	0.985 (0.926, 1.00)
RF	0.946 (0.866, 1.00)	0.846 (0.606, 1.00)	1.00 (0.921, 1.00)	0.991 (0.963, 1.00)
LG	0.973 (0.897, 1.00)	0.917 (0.736, 1.00)	1.00 (0.923, 1.00)	0.991 (0.961, 1.00)
GBDT	0.919 (0.834, 1.00)	0.833 (0.594, 1.00)	0.963 (0.873, 1.00)	0.984 (0.912, 1.00)
SVM	0.946 (0.877, 1.00)	0.900 (0.696, 1.00)	0.964 (0.903, 1.00)	0.993 (0.965, 1.00)
GNB	0.946 (0.872, 1.00)	0.829 (0.604, 1.00)	1.00 (0.938, 1.00)	0.987 (0.952, 1.00)

**Table 5 biology-12-00337-t005:** Performance of the detection models in the internal and external testing cohorts.

Models	Internal Testing Cohort (*n* = 46)				External Testing Cohort (*n* = 95)
	ACC (95% CI)	Sensitivity (95% CI)	Specificity (95% CI)	AUC (95% CI)	ACC (95% CI)
XGB	0.935 (0.890, 0.989)	0.778 (0.583, 0.912)	1.00 (0.944, 1.00)	0.982 (0.952, 1.00)	0.990 (0.883, 1.00)
RF	0.935 (0.895, 0.984)	0.778 (0.612, 0.908)	1.00 (0.939, 1.00)	0.982 (0.956, 1.00)	0.990 (0.858, 1.00)
LG	0.978 (0.967, 0.987)	0.889 (0.861, 0.913)	1.00 (0.988, 1.00)	0.991 (0.976, 1.00)	0.979 (0.824, 1.00)
GBDT	0.891 (0.836, 0.943)	0.667 (0.560, 0.813)	0.946 (0.881, 0.996)	0.928 (0.861, 0.981)	0.979 (0.796, 1.00)
SVM	0.957 (0.945, 0.988)	0.889 (0.888, 0.889)	0.973 (0.959, 1.00)	0.997 (0.992, 1.00)	0.991 (0.937, 1.00)
GNB	0.946 (0.971, 0.984)	0.889 (0.879, 0.899)	1.00 (0.992, 1.00)	0.982 (0.962, 0.996)	0.905 (0.875, 0.922)

**Table 6 biology-12-00337-t006:** Performance of the detection between radiomics-based model and radiologists in the external testing cohort.

Model	External Testing Cohort (*n* = 95)			
	Suspicious (*n*)	Diagnosed (*n*)	Missed (*n*)	ACC (95% CI)
Radiologist A	27	62	18	0.7895 (0.708–0.871)
Radiologist B	18	71	6	0.9368 (0.888–0.986)
Radiologists A and B	14	75	6	0.9368 (0.888–0.986)
Radiomics-based model	0	94	1	0.991 (0.937–1.000)
Radiologist-model diagnosis combination	6	89	0	1.00 (0.989–1.00)

ACC, accuracy; CI, confidence interval.

## Data Availability

All the data will be shared upon reasonable request by the corresponding author.

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
