# Peer review of "Diagnosis of Acute Aortic Syndromes on Non-Contrast CT Images with Radiomics-Based Machine Learning"

_biology, 2023, doi:10.3390/biology12030337_

Round 1
Reviewer 1 Report
This is an interesting testing the possibilities offered by radiomics in the identification of acute aortic syndrome (AAS) in non-contrast enhanced CT. The importance of early identification of AAS cannot be overstated, while the risk of using contrast agents in this emergency situation is not that clear. Thus, the interest in this approach is high. In this study 325 patients were enrolled and divided into a training cohort (n=135), validation cohort (n=49), internal testing cohort (n=46), and 16 external testing cohort (n=95). The number of patients with AAS in the internal cohort was 60. The Authors trained and internally and externally tested several feature selection algorithms in the diagnostic prediction of AAS.
Despite the work is interesting and has potential for improving clinical practise, there are many mayor and minors aspects that should be improved.
Mayors:
- In the setting of possible AAS the “risk of allergic reactions and renal insufficiency […] ionizing radiation and thus a risk of cancer” are likely minor risks compared to AAS. Please, consider weighting risks in light of the condition.
- Results reported in page 8 lines 219-222 and in the abstract do not match those in Table 2.
- Inter- and intra-observer reproducibility analyses are not reported. Moreover, aorta segmentations should be compared with DICE or Hausdorff distance metrics. It is unclear how ICC could be meaningful for this comparison. Reproducibility is very important here because is one of the most problematic aspects of radiomics.
- Patients enrolment methods are not clear enough. It is unclear why from 299 AAS patients available from the retrospective search only 60 were included in the internal cohort. Similar difficulties arise when considering the external cohort. Moreover, the exclusion of CT for low image quality should be detailed (now it is only included in a figure, without specifying the number of scans). Indeed, to reflect the potential clinical use, these scans should be included.
- More details about ICC computation should be given.
- “To avoid a risk of overfitting in the testing set, cross-validation was used to divide the original data into k subsets, one of which will be used as the test set and the other subsets as the training set each time”. Please, specify if this was done in subgroups of the internal training set or if included also the internal testing set.
- “stable features”. What “stable” means and how was it computed?
- The “radiomics feature selection” section is not detailed enough. Where features tested for collinearity among them without any reference to AAS detection? Was selection done sequentially, meaning ANOVA and then MUSE? How the Authors selected 542 from the 1232 available? How many feature were tested with LASSO?
- Figure 3. Was any statistical test performed to compare the distribution of these variable in AAS and NASS cohorts?
- Table 3. Please, detail in method what “suspicious” mean.
- “these artificial intelligence-powered approaches have been proven to increase the efficiency of the physician’s workflow” should be sustained by a reference.
- “As shown in Table 2, most of the classification models established with radiomics features showed good performance in the training cohort”. Table 2 does not contain results for the training cohort.
-
Minors:
- “experimental patients” is unclear.
- AAC should be introduced before first use in the Abstract. Using “accuracy” instead would facilitate the understanding.
- “In the external validation set, the ACC was 0.964 (95% CI, 0.903–1) and reached 0.991 (95% CI, 0.937–1)”. What is 0.991?
- English. “Then, radiologists A and B then decided on the interpretation together”.
- Table 1. P-values as column would be easier to identify.
- “Based on the age composition of the external validation testing cohort, the age of AAS onset in China has advanced”. This Reviewer is not sure that the inclusion criteria and search methods allow for this to be established. Please, specify or remove this sentence.
- “AD is classified according to the origin of the intimal tear or whether the dissection involves the ascending aorta (regardless of origin). The most commonly used classification schemes are the DeBakey and Stanford classifications, with the Stanford classification scheme being more commonly used in clinical practice” should be placed in “methods”, not “results”. Similarly, “Intimal calcification is age-related and present in most older adults” might be placed in discussion, and should be sustained by a reference. “First-order statistics describe the distribution of voxel intensities within the image region defined by the mask through commonly used and basic metrics” is not a result. Similarly, page 7 lines 201-207 do not contain results.
- Figure 5 is not referenced in the text.
Author Response
We are grateful for your editorial review of our article. As you have commented, there are several issues that need to be addressed. According to your valuable suggestions, we have made extensive corrections in our revised manuscript, the detailed responses are listed below. We sincerely thank the editor and reviewers for their valuable feedback that have helped to improve the quality of our manuscript. The reviewer comments are indicated below in italicized font and specific concerns have been numbered. Our response is given in normal font and changes/additions in the manuscript are given as blue text. Point 1: Referee: In the setting of possible AAS the “risk of allergic reactions and renal insufficiency […] ionizing radiation and thus a risk of cancer” are likely minor risks compared to AAS. Please, consider weighting risks in light of the condition. Reply: Thank you for reviewing our work. We reconsidered the weighting of the incidence of AAS against the incidence of iodinated contrast media (ICM) by reviewing the most recent literature. The statements have been corrected in page 2 lines 41-44. We will be happy to edit the text further based on helpful comments from the reviewers. Point 2: Referee: Results reported in page 8 lines 219-222 and in the abstract do not match those in Table 2. Reply: Thank you for reviewing our work. First of all, We sincerely apologize that we did not inform about the results of the training cohort that we included in the supporting file. Now, we have added a new Table 4 in the revised manuscript in page 8, to make the results clearer. Point 3: Referee: Inter- and intra-observer reproducibility analyses are not reported. Moreover, aorta segmentations should be compared with DICE or Hausdorff distance metrics. It is unclear how ICC could be meaningful for this comparison. Reproducibility is very important here because is one of the most problematic aspects of radiomics. Reply: Thank you for reviewing our work. As you stated, reproducibility is one of the most problematic aspects of radiomics. For aortic segmentation, we chose Dice and Hausdorff distance metrics to assess the difference in segmentation between the two physicians. Intra-class correlation coefficient (ICC) was used to evaluate the intra-observer and inter-observer agreements of radiomics feature extraction. Initially, all VOI segmentation for the 325 patients was performed by a radiologist (Radiologist A, with > 8 years of experience in vascular CT interpretation). Then 25 random CT images obtained by another radiologist (with > 10 years of experience in cardiovascular radiology) were selected to complete the above evaluation. The detailed calculation results have been added in the manuscript as attachments. Point 4: Referee: Patients enrolment methods are not clear enough. It is unclear why from 299 AAS patients available from the retrospective search only 60 were included in the internal cohort. Similar difficulties arise when considering the external cohort. Moreover, the exclusion of CT for low image quality should be detailed (now it is only included in a figure, without specifying the number of scans). Indeed, to reflect the potential clinical use, these scans should be included. Reply: Thank you for reviewing our work. We studied these patients, many of whom had a history of aortic surgery but with artifacts in their CT images that affected the diagnosis of AAS. The presence of these artifacts may affect the establishment of the model. Ultimately, we excluded these patients as well as those with incomplete images .As for the related research on 'low image quality' that you mentioned, we believe that this will be of great clinical significance, and it is also the direction of our further experiments. It is true that there are often many images with low graphics quality in life. It would be very meaningful if our model could also accurately detect these images. However, at present, our work is still ongoing, and further attention will be paid to these issues in the future. Point 5: Referee: More details about ICC computation should be given. Reply: Thank you for reviewing our work. The detailed calculation results have been added in the manuscript as attachments. Point 6: Referee: “To avoid a risk of overfitting in the testing set, cross-validation was used to divide the original data into k subsets, one of which will be used as the test set and the other subsets as the training set each time”. Please, specify if this was done in subgroups of the internal training set or if included also the internal testing set. Reply: Thank you for reviewing our work. We reexamined the data. Throughout the experiment, our data were divided as follows: 230 experimental patients (60 with AAS) were recruited from our hospital, and 95 patients with AAS were recruited as an external validation cohort from another center. The 325 patients enrolled were divided into a training cohort (n=135), validation cohort (n=49), internal testing cohort (n=46), and external testing cohort (n=95). We chose a fixed validation set, and for the established model, we completed 1000 epochs of training on the training set, and finally completed 1000 epochs of validation on the validation set as well. Point 7: Referee: “stable features”. What “stable” means and how was it computed? Reply: Thank you for reviewing our work. We apologize for not having described the term very carefully. "stable features" refer to the features that were assessed for consistency by ICC. To avoid the lack of clarity, We have changed "stable features" to "radiomic features" in page 6. Point 8: Referee: The “radiomics feature selection” section is not detailed enough. Where features tested for collinearity among them without any reference to AAS detection? Was selection done sequentially, meaning ANOVA and then MUSE? How the Authors selected 542 from the 1232 available? How many feature were tested with LASSO? Reply: Thank you for reviewing our work. We have re-written this part according to the Reviewer’s suggestion. Point 9: Referee: Figure 3. Was any statistical test performed to compare the distribution of these variable in AAS and NASS cohorts? Reply: Thank you for reviewing our work. Since the difference between AAS and NAAS groups was not clearly highlighted in Figure 3, we deleted Figure 3 and re-added Table 2, hoping to express our results more clearly. Since we included small sample data that were not normally distributed, the Mann-Whitney U test was performed to test for the difference between the two groups of data, and the results showed that there were significant statistical differences in the nine radiomics features (p<0.05). Point 10: Referee: Table 3. Please, detail in method what “suspicious” mean. Reply: Thank you for reviewing our work. We analyzed the diagnosis of the external data by two radiologists. If they accurately determined whether a patient had AAS, the result was considered a diagnosis. If the diagnosis was clearly missed, it was considered a missed diagnosis. However, if the diagnosis of AAS was suspected and the radiologists recommended for further examination, the diagnosis was regarded as suspicious. Point 11: Referee: “these artificial intelligence-powered approaches have been proven to increase the efficiency of the physician’s workflow” should be sustained by a reference. Reply: Thank you for reviewing our work. We sincerely appreciate the valuable comments. As suggested by the reviewer, we have added additional references to support this idea. Point 12: Referee: “As shown in Table 2, most of the classification models established with radiomics features showed good performance in the training cohort”. Table 2 does not contain results for the training cohort. Reply: Thank you for reviewing our work. We have made corresponding changes to address the reviewer’s second point. Minors: Point 13: Referee: “experimental patients” is unclear. Reply: Thank you for reviewing our work. A retrospective review of 325 patients who underwent aortic CT angiography in two hospitals from January 2010 to March 2022 was conducted (Fig. 1) including 230 patients from the Affiliated Hospital of Fudan University, another medical center, between January 2021 and March 2022. Point 14: Referee: AAC should be introduced before first use in the Abstract. Using “accuracy” instead would facilitate the understanding. Reply: Thank you for reviewing our work. We have revised all the nonstandard English abbreviations in the manuscript. Point 15: Referee: “In the external validation set, the ACC was 0.964 (95% CI, 0.903–1) and reached 0.991 (95% CI, 0.937–1)”. What is 0.991? Reply: Thank you for reviewing our work. We have revised and updated the manuscript to improve the description of the results. Point 16: Referee: English. “Then, radiologists A and B then decided on the interpretation together”. Reply: Thank you very much for your careful checks. We have corrected the text, and the changes can be found in page 4, line 154, in the revised manuscript. Point 17: Referee: Table 1. P-values as column would be easier to identify. Reply: Thank you for reviewing our work. We think this is an excellent suggestion. We have adjusted the layout of the table according to your comments. Point 18: Referee: “Based on the age composition of the external validation testing cohort, the age of AAS onset in China has advanced”. This Reviewer is not sure that the inclusion criteria and search methods allow for this to be established. Please, specify or remove this sentence. Reply: We think this is a very wise suggestion, and we deleted this sentence according to the opinion of the reviewer. Point 19: Referee: “AD is classified according to the origin of the intimal tear or whether the dissection involves the ascending aorta (regardless of origin). The most commonly used classification schemes are the DeBakey and Stanford classifications, with the Stanford classification scheme being more commonly used in clinical practice” should be placed in “methods”, not “results”. Similarly, “Intimal calcification is age-related and present in most older adults” might be placed in discussion, and should be sustained by a reference. “First-order statistics describe the distribution of voxel intensities within the image region defined by the mask through commonly used and basic metrics” is not a result. Similarly, page 7 lines 201-207 do not contain results. Reply: We sincerely appreciate the reviewer for the valuable review comments on our article. As you mentioned, there are several issues that should be addressed. According to your pertinent suggestions, we have made extensive corrections to improve our previous version of the manuscript. Point 20: Referee: Figure 5 is not referenced in the text. Reply: Thank you for reviewing our work. We altered and added the figure, and described the results shown in the figure in the revised manuscript again. We tried our best to improve the manuscript by revising the manuscript. These changes did not influence the content or framework of the paper. Furthermore, although we have not listed all the changes made in the revised manuscript in this response letter; however, we have shown all changes in red text in the revised paper. We sincerely appreciate for Editors/Reviewers’ valuable review and hope that the corrections meet with your approval.

Reviewer 2 Report
The authors of this study suggested a machine learning approach based on radiomics to identify acute aortic syndromes (AAS) in non-contrast CT images. 325 patient cases in all were examined. Nine feature parameters were identified using the Least Absolute Shrinkage and Selection Operator regression methods: (original_shape_SurfaceVolumeRatio, original_firstorder_Maximum, original_glcm_Imc1, original_shape_MeshVolume, original_glrlm_LongRunHighGrayLevelEmphasis, wavelet.LLH_firstorder_Mean, wavelet.LLH_glcm_Correlation, wavelet.LHL_gldm_GrayLevelNonUniformity, and wavelet.HLH_glszm_GrayLevelNonUniformity). Then, eight machine-learning models were tested (XGBoost, Random Forest, Logistic Regression, GBDT, Decision Tree, K-NN, GaussianNB and SVM). The SVM model demonstrated the best results, according to the authors. It can identify AAS on unenhanced CT, minimizing missed and incorrect diagnoses.
The paper may be interesting, however there are several grammatical errors, and it lacks structure and clarity. Significant changes are required.
The motivation for the suggested novelty is not stated in the introduction. The problem's introduction needs to be reorganized; thus, the highlighted research issues should be thoroughly mapped in contributions.
Regarding the database, the authors stressed that the Helsinki Accord was adhered to during the data gathering, which began in January 2010. The thickness and size of the slices, as well as the scanner used, are all carefully described by the authors. The aorta volume's manual segmentation as well as inter- and intra-expert variability were also introduced. But authors don't mention whether or not their database will be made public? If this work is ever published, I strongly advise that this database be made anonymously available to enable for potential comparisons in the future.
Despite being extensively comprehensive, the flowchart of base collection and model creation lacks visual illustrations. For a better understanding, including certain 2D cuts or 3D renderings, even in miniature, can be very beneficial.
References must be made to each of the models listed under "Construction of the radiomic signature" section. These models have been used in hundreds of works, particularly those involving medical imaging.
The authors discuss the techniques and architectures utilized in the statistical analysis part without explaining why. For example, are all models stable after 1000 epochs?
Also, it seems wise to reformulate this component in relation to the graphic card's settings. In fact, stating that an RTX 30xx is required actually imply that no other platform can execute the code (RTX 3090, Colab)? It would be more accurate to say that running models require a minimum of xx CPU, xx RAM, and RTX 30xx.
The section on "Patient characteristics" seems to hold up. However, it's still unclear how to get from 1232 to 9 features. I keep going back to the “why” issue. Why not stick to just these nine traits, and under what scenario?
Unfortunately, the article only includes a portion of figures 2 and 3. They should be scaled down. Please check for readability before submitting the new version.
Although very helpful, the parameters introduced in table 2 are now quite significant. A few of models exceed 0.9 ACC ? For other models, the measured sensitivity is similarly close to 0.9. At least three models—Logistic Regression, SVM, and Gaussian NB—are comparable to one another for an external reader. The use of SVM as a reference model has to be more thoroughly explained by the authors. It is crucial to stress that while processing medical images, it is highly recommended to construct a graph that displays how well each model performs across all classification thresholds. The true positive rate and false positive rate, which are represented by the ROC curve (Receiver Operating Characteristic), will enable a better understanding and identification of the optimal model.
The discussion section may need some work. Since it must be acknowledged that neither the architecture nor the SVM's settings have not been changed, it is not only important to compare the performances of the models in the literature, but also to highlight the authors' contributions, which go beyond a straightforward application of the basic model.
The bibliography must be updated and consolidated with new publications. Of the twenty references given by the authors, more than half are published before 2020.
Author Response
We are grateful for your editorial review of our article. As you have commented, there are several issues that need to be addressed. According to your valuable suggestions, we have made extensive corrections in our revised manuscript, the detailed responses are listed below. We sincerely thank the editor and reviewers for their valuable feedback that have helped to improve the quality of our manuscript. The reviewer comments are indicated below in italicized font and specific concerns have been numbered. Our response is given in normal font and changes/additions in the manuscript are given as blue text.
Point 1:
Referee: The motivation for the suggested novelty is not stated in the introduction. The problem's introduction needs to be reorganized; thus, the highlighted research issues should be thoroughly mapped in contributions.
Reply: Thank you for reviewing our work. We consider these comments as good suggestions. We have revised the Introduction section in the manuscript in accordance with the comments of the reviewing experts.
Point 2:
Referee: Regarding the database, the authors stressed that the Helsinki Accord was adhered to during the data gathering, which began in January 2010. The thickness and size of the slices, as well as the scanner used, are all carefully described by the authors. The aorta volume's manual segmentation as well as inter- and intra-expert variability were also introduced. But authors don't mention whether or not their database will be made public? If this work is ever published, I strongly advise that this database be made anonymously available to enable for potential comparisons in the future.
Reply: Thank you for reviewing our work. We are willing to host a competition on aortic dissection rupture identification, and we will then provide corresponding data on the competition in the future.
Point 3:
Referee: Despite being extensively comprehensive, the flowchart of base collection and model creation lacks visual illustrations. For a better understanding, including certain 2D cuts or 3D renderings, even in miniature, can be very beneficial.
Reply: Thank you for reviewing our work. We really appreciate your comments and have modified the illustrations in the manuscript.
Point 4:
Referee: References must be made to each of the models listed under "Construction of the radiomic signature" section. These models have been used in hundreds of works, particularly those involving medical imaging.
Reply: Thank you for reviewing our work. As suggested by the reviewer, we have added additional references to support this idea.
Point 5:
Referee: The authors discuss the techniques and architectures utilized in the statistical analysis part without explaining why. For example, are all models stable after 1000 epochs?Also, it seems wise to reformulate this component in relation to the graphic card's settings. In fact, stating that an RTX 30xx is required actually imply that no other platform can execute the code (RTX 3090, Colab)? It would be more accurate to say that running models require a minimum of xx CPU, xx RAM, and RTX 30xx.
Reply: Thank you for reviewing our work. To choose a stable model parameter, we set the epoch parameter to 1000 and our training environment was done in the graphics processing unit (GPU, RTX 3090) environment. Deep learning, the classification model based on radiomics data, does not require the support of GPU; moreover, our model can also run in the central processing unit (CPU) environment.
Point 6:
Referee: The section on "Patient characteristics" seems to hold up. However, it's still unclear how to get from 1232 to 9 features. I keep going back to the “why” issue. Why not stick to just these nine traits, and under what scenario?
Reply: Thank you for reviewing our work. We have re-written this part, according to the Reviewer’s suggestion.
Point 7:
Referee: Unfortunately, the article only includes a portion of figures 2 and 3. They should be scaled down. Please check for readability before submitting the new version.
Reply: Thank you for reviewing our work. We sincerely appreciate the reviewer for the valuable review comments on our article. As you mentioned, there are several issues that should be addressed. According to your pertinent suggestions, we have made extensive corrections to improve our previous version of the manuscript.
Point 8:
Referee: Although very helpful, the parameters introduced in table 2 are now quite significant. A few of models exceed 0.9 ACC ? For other models, the measured sensitivity is similarly close to 0.9. At least three models—Logistic Regression, SVM, and Gaussian NB—are comparable to one another for an external reader. The use of SVM as a reference model has to be more thoroughly explained by the authors.
Reply: Thank you for reviewing our work. We have updated the manuscript with more detailed results, and we have added Table 3
Point 8:
It is crucial to stress that while processing medical images, it is highly recommended to construct a graph that displays how well each model performs across all classification thresholds. The true positive rate and false positive rate, which are represented by the ROC curve (Receiver Operating Characteristic), will enable a better understanding and identification of the optimal model.
Reply: Thank you for reviewing our work. We have added the ROC curve results in the manuscript (Figure 3)
Point 9:
Referee: The discussion section may need some work. Since it must be acknowledged that neither the architecture nor the SVM's settings have not been changed, it is not only important to compare the performances of the models in the literature, but also to highlight the authors' contributions, which go beyond a straightforward application of the basic model.
Reply: Thank you for reviewing our work. We very much agree with your comments, which are very helpful to us, and we have revised the discussion section in the manuscript accordingly.
Point 10:
Referee: The bibliography must be updated and consolidated with new publications. Of the twenty references given by the authors, more than half are published before 2020.
Reply: Thank you for reviewing our work. We sincerely appreciate the valuable comments. We have checked the literature carefully and added additional and more recent references in the revised manuscript.

Round 2
Reviewer 1 Report
This reviewer thanks the Authors for considering these comments. While the manuscript improved, there are still a number of issued.
With no information on the failure of the model/technique on images of low quality it is impossible to evaluate whether this model could be used in clinical practise.
English should be revised. Examples “Total 230 experimental patients 12 (60 with AAS) were recruited from our hospital”, “Entire aorta was marked by a radiologist”, “Figure 3. most of the”, “the radiologists assisted in the diagnosis of our SVM model,”, “These NCCT images without significant characteristics”, “features is more convincing than that of a”,
The Authors also failed to explain how cross-validation was used.
The internal testing cohort only included 9 patients with AAS. Moreover, the lack of non AAS cases in the external validation cohort precludes the computation of any descriptors of model performances beyond ACC. Non AAS cases from the external hospital are required to evaluate the model and to sustain the “detection performance”, both for the radiologists and for the models.
What was the unit of measure of HF95? It is impossible to understand the results without knowing it.
It is unclear how inter-observer reproducibility was obtained. In the method there is no reference to it.
“shown in Table 5, when only the primary doctor was on duty in the ED,”. I understand that these tests were not performed when “on duty in the ED” but later on in a research setting. Please, make it clear.
Figure 3 and 5 are impossible to read.
Considering the very low performance of KNN and decision tree model (and others) and the overall complexity of the manuscript, information on these models and their results could be moved to supplementary material. What is the added information of showing so many different model, if anyway just one is chosen?
Author Response
Dear reviewer:
We feel great thanks for your professional review work on our article. As you are concerned, there are several problems that need to be addressed. According to your nice suggestions, we have made extensive corrections to our previous draft, the detailed corrections are listed below. We sincerely thank the editor and all reviewers for their valuable feedback that we have used to improve the quality of our manuscript. The reviewer comments are laid out below in italicized font and specific concerns have been numbered. Our response is given in normal font and changes/additions to the manuscript are given in the blue text. If there are any other modifications we could make, we would like very much to modify them and we really appreciate your help.
Point 1:
Referee: With no information on the failure of the model/technique on images of low quality it is impossible to evaluate whether this model could be used in clinical practise.
Reply: Thank you for reviewing our work. We sincerely thank the reviewers for their valuable comments. As you mentioned in your first amendment: " Moreover, the exclusion of CT for low image quality should be detailed (now it is only included in a figure, without specifying the number of scans). Indeed, to reflect the potential clinical use, these scans should be included".
I am sorry that we did not describe it clearly in the manuscript. We believe that our model has great clinical significance. On the one hand, we excluded low-quality images (such as the presence of image artifacts, after surgery), most of these patients were after the implantation of artificial blood vessels and stent vessels, which are easily identified by human experts on the NCCT. Our model aims to identify patients who are difficult for human experts to diagnose AAS on the NCCT, and to enable the screening of patients with potential AAS. On the other hand, our cases were collected at two independent medical centers, and there were differences in slice thickness between the training cohort and the external testing cohort. On the training cohort, we used high-resolution CT with slice thickness around 1.5mm. However, for the external testing cohort, our images were all images with a slice thickness of 5mm. Despite the differences in image quality, our model performed well on both the training and external testing cohort. Therefore, we believe that such a model has great clinical value. We have added this information to the manuscript in the hope of gaining your approval.
Revision:{Page 2, line 85} We excluded low-quality images from patients with prosthetic vessels and stents, who were readily diagnosed by a human expert on the NCCT.
Point 2:
Referee: English should be revised. Examples “Total 230 experimental patients 12 (60 with AAS) were recruited from our hospital”, “Entire aorta was marked by a radiologist”, “Figure 3. most of the”, “the radiologists assisted in the diagnosis of our SVM model,”, “These NCCT images without significant characteristics”, “features is more convincing than that of a”,
Reply: We apologize for the poor language of our manuscript. We worked on the manuscript for a long time and the repeated addition and removal of sentences and sections obviously led to poor readability. We have now worked on both language and readability and have also involved native English speakers for language corrections. We really hope that the flow and language level have been substantially improved.
Original article:{Page 1, line 12}Total 230 experimental patients (60 with AAS) were recruited from our hospital, and 95 patients with AAS were recruited as an external testing cohort.
Revision:A total of 325 patients who underwent aortic CT angiography (CTA) were enrolled retrospectively from two medical centers in China to form the internal cohort (230 patients, 60 patients with AAS) and the external testing cohort (95 patients with AAS). The internal cohort was divided into the training cohort (n = 135), validation cohort (n = 49), and internal testing cohort (n = 46).
Original article:{Page 1, line 16} Entire aorta was marked by a radiologist on all images
Revision:The aortic mask was manually delineated on NCCT by a radiologist.
Original article:{Page 7, line 250}Figure 3. most of the
Revision:Most of the classification models established with radiomics features showed good performance in the validation cohort.
Original article:{Page 9, line 291}“the radiologists assisted in the diagnosis of our SVM model,”
Revision: {Page 9, line 281}As shown in Table 6, the occurrence of AAS on the NCCT images of the external testing cohort was interpreted independently by one radiologist, and the rate of missed diagnosis of AAS was 0.189. Six patients with AAS were missed despite review by senior radiologists. Our SVM radiomics model was able to accurately detect AAS patients in the external testing cohort. Similarly, patients with AAS who presented nonspecific on the NCCT were more likely to be identified when radiologists used our radiomics SVM model to assist diagnosis.
Original article:{Page 9, line 301} “These NCCT images without significant characteristics”
Revision:These NCCT images are devoid of distinctive features (e.g., displaced calcified intimal flaps, intraluminal linear high density, intramural hematoma, and aneurysmal dilatation). These features are very evident on CTA. None of the four radiologists made an accurate diagnosis on the NCCT, and all six patients were missed.
Original article:{Page 12, line 494} “features is more convincing than that of a”
Revision:We have removed the ambiguous sentence.
Point 3:
Referee: The Authors also failed to explain how cross-validation was used.
Reply: Thanks for your careful checks. We are sorry for our carelessness. Figure 1 illustrates the flow chart for patient enrollment and model construction. We chose the set-aside method to divide the experimental data. The 325 patients enrolled were divided into a training cohort (n=135), validation cohort (n=49), internal testing cohort (n=46), and external testing cohort (n=95).
Point 4:
Referee: The internal testing cohort only included 9 patients with AAS. Moreover, the lack of non AAS cases in the external validation cohort precludes the computation of any descriptors of model performances beyond ACC. Non AAS cases from the external hospital are required to evaluate the model and to sustain the “detection performance”, both for the radiologists and for the models.
Reply:
Thank you very much for pointing out this important issue. We agree with your comments that preliminary experiments are also necessary. Your suggestion provides a direction for our next research.
This study aimed to investigate the accurate diagnosis of AAS on the NCCT. Therefore, we focused on the detection of AAS. A recent American study found an incidence of 7.7 per 100,000 person-years for AAD, IMH, and PAU, in which the incidence of AAD was 4.4 per 100,000 person-years. Therefore, we believe that it is more important to collect images of AAS patients. We trained a radiomics model on internal data that could accurately identify AAS. Therefore, in order to better validate the model we established, we used the NCCT images of all AAS patients in the external validation dataset.
Point 5:
Referee: What was the unit of measure of HF95? It is impossible to understand the results without knowing it.
Reply: Thank you for your careful review of our work. I'm sorry we didn't explain this in detail.
First of all, I am sorry that we did not properly abbreviate proper nouns, which caused you to have doubts when reading our manuscript. We have added an explanation of HD95 and its related references to the manuscript.
Revision:{Page 4, line 129}95%Hausdorff distance(HD95) is a metric of using Hausdorff distance to measure the 95% quantile of the surface distance. HD95 was used to evaluate the degree of matching between manually segmented images and Ground Truth, and the unit of measurement was mm. The larger the HD95, the higher the mismatch between the two images(1).
- Dongwei,L.; Ning,S.; Tao,H.; Wei,W.; Jianxia,Z.; Jianxin,Z. SGEResU-Net for brain tumor segmentation. Math Biosci Eng 2022, 30, 5576-5590
Point 6:
Referee: It is unclear how inter-observer reproducibility was obtained. In the method there is no reference to it.
Reply: Dear editors and reviewers, we describe in our methods how to control interobserver repeatability. We have modified this in the manuscript.
Revision:{Page 4, line 128} Dice and 95%Hausdorff distance(HD95) were used to evaluate the differences in aortic segmentation between radiologists. HD95 is a metric of using Hausdorff dis-tance to measure the 95% quantile of the surface distance. HD95 was used to evaluate the degree of matching between manually segmented images and Ground Truth, and the unit of measurement was mm. The larger the HD95, the higher the mismatch between the two images. Intraclass correlation coefficient (ICC) was used to evaluate the reliability in radiomics values between the two radiologists. ICC is the reliability coefficient measuring and evaluating inter-observer reliability and test-retest reliability. In our study, ICC were calculated based on a single-measurement, absolute-agreement, two-way random-effects model. Originally, all VOI segmentations in the 325 patients were performed by one radiologist (radiologist A, with > 8 years of experience in vascular CT interpretation). Then, 30 random CT images obtained by another radiologist (with > 10 years of experience in cardiovascular radiology) were selected for the evaluation of reliability, detailed in Supplementary S1.
Point 7:
Referee: “shown in Table 5, when only the primary doctor was on duty in the ED,”. I understand that these tests were not performed when “on duty in the ED” but later on in a research setting. Please, make it clear.
Reply: Thank you for your critical comments and we totally agree with your suggestions which might be of great help to improve the quality of our manuscript.According to the reviewer’s suggestions, we performed some modifications in our manuscript.
Revision:{Page 9, line 291}As shown in Table 6, the occurrence of AAS on the NCCT images of the external testing cohort was interpreted independently by one radiologist, and the rate of missed diagnosis of AAS was 0.189. Six patients with AAS were missed despite review by senior radiologists. Our SVM radiomics model was able to accurately detect AAS patients in the external testing cohort. Similarly, patients with AAS who presented nonspecific on the NCCT were more likely to be identified when radiologists used our radiomics SVM model to assist diagnosis.
Point 8:
Referee: Figure 3 and 5 are impossible to read.
Reply: According to the editor and reviewers’ comments, we have made extensive modifications to our manuscript to make our results convincing. We replaced the illustrations and added detailed illustration notes to the manuscript. Thank you again for your positive comments and valuable suggestions to improve the quality of our manuscript.
Revision:
Figure 3. AAS detection performance results of the models in the training, validation, and exter-nal test cohorts. ACC, accuracy;AUC, area under the curve ; XGB, eXtreme Gradient Boostin; RF, Random Forest; LG, logistic regression; GBDT, Gradient Boosting Decision Tree; SVM, Support Vector Machine; GNB, GaussianNB.
Figure 5. Radiomic features of the six patients diagnosed with AAS.(A) original_shape_Surface_Volume Ratio. (B) Wavelet_LLH_firstorder_Mean. (C) original_gldm_GrayLevelNonUniformity. (D) original_glrlm_LongRunHighGrayLevelEmphasis.
Point 9:
Referee: Considering the very low performance of KNN and decision tree model (and others) and the overall complexity of the manuscript, information on these models and their results could be moved to supplementary material. What is the added information of showing so many different model, if anyway just one is chosen?
Reply: Thank you very much for your careful review of our manuscript and constructive suggestions. At the same time, I would also like to thank you for your specific modification strategy. I have made comprehensive and detailed modifications according to your suggestion.
Revision:Given the low performance of KNN and decision tree models and the overall complexity of the manuscript, information about these models and their results has been transferred to the supplementary material(Supplementary S2).

Reviewer 2 Report
The necessary corrections have been made by the authors. As indicated, the article has been updated. I support the publication of this work. I continue to urge that the database be made available as supplementary resource.
Author Response
Dear reviewer:
We feel great thanks for your professional review work on our article.
Point 1:
Referee: The necessary corrections have been made by the authors. As indicated, the article has been updated. I support the publication of this work. I continue to urge that the database be made available as supplementary resource.
Reply: Thank you for reviewing our work. We think this is a good suggestion. Thank you very much for your hard work!We will carry out this work as soon as possible.
Round 3
Reviewer 1 Report
I would like to thank the Authors for the consideration of my previous comments. I have no more comments or critics to add.